# Modulation of anxiety and fear via distinct intrahippocampal circuits

**Elif Engin[1,2]\*, Kiersten S Smith[1,2], Yudong Gao[3], David Nagy[4], Rachel A Foster[1,2], Evgeny Tsvetkov[2,5,6], Ruth Keist[7], Florence Crestani[7], Jean-Marc Fritschy[7], Vadim Y Bolshakov[2,5], Mihaly Hajos[4], Scott A Heldt[3], Uwe Rudolph[1,2]**

[1]Laboratory of Genetic Neuropharmacology, McLean Hospital, Belmont, United States; [2]Department of Psychiatry, Harvard Medical School, Boston, United States; [3]Department of Anatomy and Neurobiology, The University of Tennessee Health Science Center, Memphis, United States; [4]Section of Comparative Medicine, Yale School of Medicine, New Haven, United States; [5]Cellular Neurobiology Laboratory, McLean Hospital, Belmont, United States; [6]Sechenov Institute of Evolutionary Physiology and Biochemistry, Russian Academy of Sciences, St. Petersburg, Russia; [7]Institute for Pharmacology and Toxicology, University of Zurich, Zurich, Switzerland

**Abstract** Recent findings indicate a high level of specialization at the level of microcircuits and cell populations within brain structures with regards to the control of fear and anxiety. The hippocampus, however, has been treated as a unitary structure in anxiety and fear research despite mounting evidence that different hippocampal subregions have specialized roles in other cognitive domains. Using novel cell-type- and region-specific conditional knockouts of the $GABA_A$ receptor $\alpha2$ subunit, we demonstrate that inhibition of the principal neurons of the dentate gyrus and CA3 via $\alpha2$-containing $GABA_A$ receptors ($\alpha2GABA_ARs$) is required to suppress anxiety, while the inhibition of CA1 pyramidal neurons is required to suppress fear responses. We further show that the diazepam-modulation of hippocampal theta activity shows certain parallels with our behavioral findings, suggesting a possible mechanism for the observed behavioral effects. Thus, our findings demonstrate a double dissociation in the regulation of anxiety versus fear by hippocampal microcircuitry.

*For correspondence: eengin@mclean.harvard.edu

## Introduction

Fear and anxiety are distinct emotional states induced by different environmental triggers (acute, objectively harmful stimuli versus the possibility of unidentifiable, obscure threats, respectively) and resulting in distinguishable defensive behaviors (freezing, fight or immediate active avoidance versus alertness and risk-assessment [*Tovote et al., 2015*; *Davis et al., 2010*]). Recent circuit-focused studies demonstrate that the neurocircuitry and neuronal cell populations mediating fear and anxiety show some overlap (*Botta et al., 2015*; *Jennings et al., 2013*), but also significant divergence (*Kheirbek et al., 2013*; *Yamaguchi et al., 2013*). Here, we investigated whether fear and anxiety converge or diverge at the level of microcircuits within the hippocampus (HPC).

The intra-HPC circuitry includes three subregions (CA1 and CA3 areas, and dentate gyrus (DG)), with predominantly unidirectional excitatory projections from DG to CA3 to CA1. Additionally, the principal neurons in each subregion are tightly regulated via activity of $GABA_A$ receptors ($GABA_ARs$). Out of the five $GABA_AR$ subtypes expressed in the HPC, the $\alpha2$-containing $GABA_ARs$ ($\alpha2GABA_AR$) have been strongly and consistently implicated in anxiety and fear (*Vollenweider et al., 2011*; *Löw et al., 2000*; *Smith et al., 2012*). $\alpha2GABA_ARs$ are expressed on principal neurons in all three

**eLife digest** Fear and anxiety can be thought of as different but related emotional states. Fear is triggered by specific harmful situations, such as the immediate presence of a predator. Anxiety instead results from the possibility of an obscure threat, such as being in an exposed environment, which increases the chance of being detected by a predator. Evidence suggests that slightly different areas of the brain control fear and anxiety, but much remains unknown about the specific brain regions that help to regulate these two emotional states.

One brain region that has been implicated in both anxiety and fear – as well as in learning and memory – is the hippocampus. Named after the Greek word for seahorse because of its shape, the hippocampus is made up of three subregions: CA1, CA3 and the dentate gyrus. Each of these subregions has a distinct role in learning and memory. However, their individual contributions to the control of fear and anxiety were not known.

An inhibitory receptor protein found in the surface of some hippocampal neurons had previously been shown to be involved in controlling fear and anxiety. Now, Engin et al. have studied three different groups of genetically modified mice, each of which lacks the receptor protein in a different subregion of the hippocampus. The mice completed tests that stimulated anxiety or fear, some while under the influence of the anxiety and fear-reducing drug diazepam. Notably, diazepam failed to reduce fear in animals that lacked the inhibitory receptor protein in the CA1 subregion of the hippocampus, suggesting that this subregion participates in the fear response. However, mice that lacked the receptor in the dentate gyrus or CA3 responded normally to the drug (they showed reduced fear when given diazepam).

In tests of anxiety, the picture was exactly the opposite. Diazepam failed to reduce anxiety in animals lacking the inhibitory receptor in the dentate gyrus or CA3, indicating that these subregions are involved in the regulation of anxiety. However, the drug still reduced anxiety in mice that lacked the receptor protein in the CA1 subregion.

Further studies are now needed to clarify how manipulating specific subregions of the hippocampus alters how it communicates with other brain structures to generate changes in anxiety or fear-related behaviors.

HPC subregions and mediate fast phasic inhibition (*Fritschy and Mohler, 1995*; *Hörtnagl et al., 2013*).

To address the question of how HPC microcircuits contribute to anxiety versus fear, we generated three gene-targeted mouse lines in which the $\alpha$2GABA$_A$Rs are deleted selectively in the pyramidal neurons of the CA1 or of the CA3, or in the granule cells of the DG. Reducing the expression of synaptic GABA$_A$Rs later than approximately 2–3 weeks postnatally typically causes no baseline anxiety phenotype (*Earnheart et al., 2007*; *Shen et al., 2012*), in line with the concept of a developmental origin of anxiety (*Gross et al., 2002*; *Gross and Hen, 2004*). The cre driver lines used in the current study express cre recombinase later in development, and our conditional knockout mice display no spontaneous anxiety or fear phenotype. To assess which part of intrahippocampal circuitry is essential for modulation of anxiety and fear, we thus used a mixed genetic-pharmacological approach: We systemically administered a nonselective positive allosteric modulator of GABA$_A$ receptors, known to induce anxiolysis and fear reduction, in cell type- and region-specific $\alpha$2 knockout mice. Our findings demonstrate a double-dissociation within the HPC with regards to fear and anxiety, where inhibition of CA3 and DG is required for anxiolysis, while the inhibition of CA1 is required for the reduction of fear.

## Results

### Basic characterization of HPC conditional knockout mice

A floxed *Gabra2* allele was generated by placing two *lox*P sites 1 kb apart flanking exon 5 (221 bp) (see *Witschi et al., 2011* for details; *Figure 1A*). The $\alpha$2 conditional knockout mice were generated by crossing mice homozygous for the floxed *Gabra2* allele ($\alpha$2F/F mice) with $\alpha$2F/F mice carrying

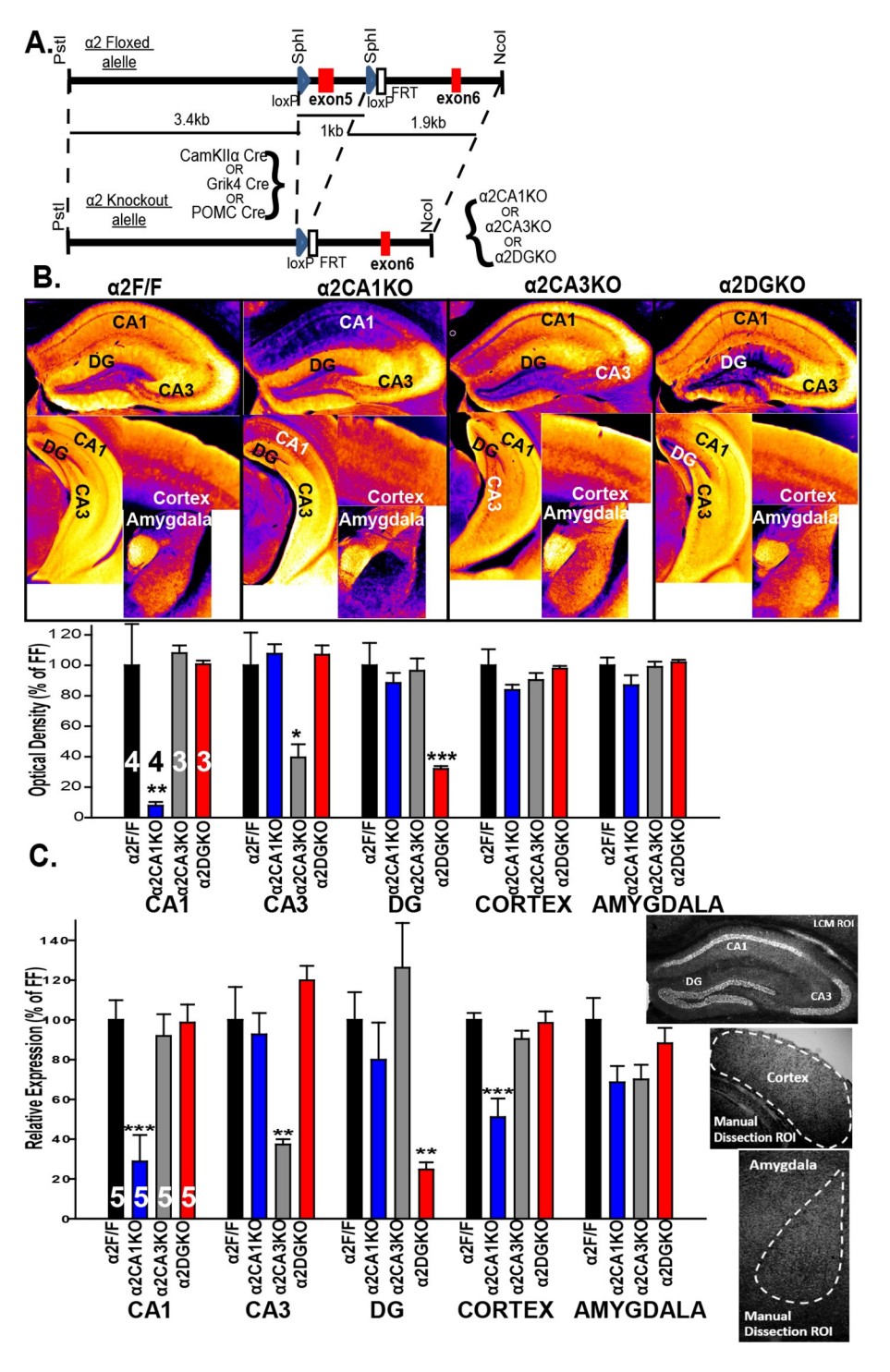

**Figure 1.** Targeted reduction of α2 expression in CA1, CA3 or DG. (**A**) Generation of the α2F/F control and α2CA1KO, α2CA3KO, α2DGKO mice. (**B**) Top: False color images showing the α2 staining intensity in immunohistochemically stained sections. Cooler colors = less staining. Bottom: Semi-quantitative comparisons of α2 staining. (**C**) α2 mRNA expression (see sample ROI's on the right), expressed as% of α2F/F control.*p<0.05, **p<0.01, ***p<0.001 compared to corresponding α2F/F group.

The following figure supplements are available for figure 1:

*Figure 1 continued on next page*

*Figure 1 continued*

**Figure supplement 1.** Immunohistochemical localization of GABA$_A$R $\alpha$2 subunits in conditional knockout mice bred on a 129X1/SvJ background.

**Figure supplement 2.** Expression of $\alpha$1 and $\alpha$5 subunits in conditional knockout mice.

**Figure supplement 3.** Expression of $\alpha$3 and $\alpha$4 mRNA in conditional knockout mice.

**Figure supplement 4.** Miniature inhibitory postsynaptic currents.

**Figure supplement 5.** Tests of hippocampal function.

one of the following three cre recombinase transgenes: CaMKII$\alpha$ cre (T29-1 mice; *Tsien et al., 1996*) to generate a CA1 pyramidal neuron selective knockout ($\alpha$2CA1KO), Grik4 cre (G32-4 mice; *Nakazawa et al., 2002* ) to generate a CA3 pyramidal neuron selective knockout ($\alpha$2CA3KO) and POMC cre (*McHugh et al., 2007*) to generate a DG granule cell selective knockout ($\alpha$2DGKO).

Within the HPC, the reduction in $\alpha$2 expression was limited to the targeted regions at both the protein (*Figure 1B*; *Figure 1—figure supplement 1*) and mRNA (*Figure 1C*) level for each genotype (*Table 1*, Sect. 1, 2). The knockdowns extended through the septo-temporal axis of the hippocampus, although they were more pronounced in dorsal regions (*Figure 1B*; *Figure 1—figure supplement 1*). $\alpha$2 mRNA expression was also reduced in the cortex of $\alpha$2CA1KO mice, but this reduction was not observed at the protein level at the time point examined (10–11 weeks). CaMKII$\alpha$ cre-mediated recombination is progressive over time in cortex (*Fukaya et al., 2003*), and the mRNA expression might be already reduced at this time point and this reduction may start to be reflected at the functional protein level later in development. The expression of other GABA$_A$R subunits in the regions of interest was largely unaffected (*Figure 1*, *Figure 1—figure supplement 2*, *Figure 1—figure supplement 3*; *Table 1*).

The conditional knockouts of the *Gabra2* gene did not cause major changes in the frequency, amplitude or decay kinetics of the miniature inhibitory postsynaptic currents (mIPSCs), and did not impair the response to diazepam in any of the genotypes (*Figure 1—figure supplement 4*; *Table 1*, Sect. 3). Cognitive tests of hippocampal function revealed no gross anomalies in any of the knockouts (*Figure 1—figure supplement 5*; *Table 1*, Sect. 4). Thus, the knockout of the *Gabra2* gene in CA1, CA3 or DG was limited to the targeted regions and the targeted receptor, and did not cause major impairments in baseline inhibitory synaptic activity or the diazepam-induced changes of inhibitory synaptic responses in principal neurons of the targeted regions in the hippocampus.

## Behavioral tests of anxiety

Anxiety-related behavior was measured using two validated tests of anxiety (*Treit et al., 2010*): Elevated plus maze (EPM) and light/dark box (LDB). In EPM, enhancing GABA$_A$R-mediated responses with diazepam increased open arm activity in $\alpha$2F/F and $\alpha$2CA1KO, but not in $\alpha$2CA3KO or $\alpha$2DGKO mice (*Figure 2A*, *Figure 2—figure supplement 1A*; *Table 2*, Sect. 1). While the diazepam effects were clear in $\alpha$2F/F and $\alpha$2CA1KO mice, the lack of these effects especially in $\alpha$2CA3KO mice was partially due to an increase in percent open arm time in the vehicle condition, as well as a decrease in the same measure in the diazepam condition. To clarify these findings, we repeated this test with mice this time bred on a 129X1/SvJ background under slightly different testing conditions (see Methods; the conditional knockouts bred on the 129X1/SvJ background show similar distribution of GABA$_A$R $\alpha$2 subunits as the C57BL/6J; *Figure 1—figure supplement 1*). Differences between vehicle-treated groups were smaller in this strain, and yet the pattern of diazepam effects were identical to those reported with the C57BL/6J strain, with significant anxiolytic-like effects in $\alpha$2F/F and $\alpha$2CA1KO, but not in $\alpha$2CA3KO or $\alpha$2DGKO mice (*Figure 2—figure supplement 1B*; *Table 2*, Sect. 1').

LDB experiments were conducted only on mice bred on the 129X1/SvJ background, as diazepam did not lead to consistent anxiolytic-like effects in control mice of C57BL/6J background. Similar to EPM, diazepam increased the time animals spent in the larger lit compartment of the LDB in $\alpha$2F/F

**Table 1.** Results of omnibus statistical tests of experiments for the general characterization of $\alpha$2CA1KO, $\alpha$2CA3KO and $\alpha$2DGKO mice.

#### 1. Immunohistochemistry

One-Way ANOVA; Factor: Genotype

|  | $\alpha$2 Subunit | | $\alpha$1 Subunit | | $\alpha$5 Subunit | |
|---|---|---|---|---|---|---|
| CA1 | F(3,10)=9.44 | P=0.003 | F(3,9)=0.08 | p=0.97 | F(3,9)=0.10 | p=0.96 |
| CA3 | F(3,10)=5.05 | P=0.02 | F(3,9)=0.01 | p=0.99 | F(3,9)=0.13 | p=0.94 |
| DG | F(3,10)=9.08 | P=0.003 | F(3,9)=0.00 | p=1.00 | F(3,9)=0.06 | p=0.98 |
| Cortex | F(3,10)=1.90 | p=0.19 | F(3,9)=0.01 | p=0.83 | F(3,9)=0.01 | p=0.99 |
| Amygdala | F(3,10)=1.28 | p=0.34 | F(3,9)=0.21 | p=0.89 | F(3,9)=0.36 | p=0.78 |

#### 2. Quantitative PCR

One-Way ANOVA; Factor: Genotype

|  | $\alpha$2 Subunit | | $\alpha$3 Subunit | | $\alpha$4 Subunit | |
|---|---|---|---|---|---|---|
| CA1 | F(3,16)=10.66 | p<0.001 | F(3,16)=0.10 | p=0.96 | F(3,16)=2.48 | p=0.1 |
| CA3 | F(3,16)=10.53 | p<0.001 | F(3,16)=0.67 | p=0.58 | F(3,16)=4.74 | p=0.02 |
| DG | F(3,16)=7.32 | p=0.003 | F(3,12)=1.96 | p=0.17 | F(3,12)=2.90 | p=0.08 |
| Cortex | F(3,16)=15.69 | p<0.001 | | | | |
| Amygdala | F(3,16)=3.01 | p=0.06 | | | | |

#### 3. Slice Electrophysiology

Two-Way Mixed Factorial ANOVA; Factors: Genotype (between-subjects), Drug (within-subjects)

| CA1 | Amplitude | | Frequency | | Decay Time | |
|---|---|---|---|---|---|---|
| Genotype | F(1,38)=1.83 | p=0.18 | F(1,38)=2.80 | p=0.10 | F(1,38)=1.94 | p=0.17 |
| Drug | F(1,38)=1.49 | p=0.23 | F(1,38)=9.38 | p=0.004 | F(1,38)=107.96 | p<0.001 |
| Genotype x Drug Interaction | F(1,38)=2.91 | p=0.09 | F(1,38)=3.11 | p=0.08 | F(1,38)=0.82 | p=0.37 |
| CA3 | Amplitude | | Frequency | | Decay Time | |
| Genotype | F(1,21)=2.36 | p=0.14 | F(1,21)=1.85 | p=0.19 | F(1,21)=0.95 | p=0.34 |
| Drug | F(1,21)=0.66 | p=0.42 | F(1,21)=0.09 | p=0.77 | F(1,21)=30.54 | p<0.001 |
| Genotype x Drug Interaction | F(1,21)=0.21 | p=0.65 | F(1,21)=1.90 | p=0.18 | F(1,21)=0.25 | p=0.62 |
| DG | Amplitude | Frequency | Decay Time | | | |
| Genotype | F(1,21)=1.50 | p=0.23 | F(1,21)=1.58 | p=0.22 | F(1,21)=0.01 | p=0.91 |
| Drug | F(1,21)=2.58 | p=0.12 | F(1,21)=1.58 | p=0.22 | F(1,21)=47.42 | p<0.001 |
| Genotype x Drug Interaction | F(1,21)=0.54 | p=0.47 | F(1,21)=1.33 | p=0.26 | F(1,21)=0.30 | p=0.59 |

#### 4. Tests of Hippocampal Function

Delay – Trace Fear Conditioning

Two-Way Factorial ANOVA; Factors: Genotype (between-subjects), Condition (between-subjects)

|  | % Freezing | |
|---|---|---|
| Genotype | F(3,49)=0.38 | p=0.77 |
| Condition | F(1,49)=41.57 | p<0.001 |
| Genotype x Cond. Interaction | F(3,49)=0.71 | p=0.55 |

Contextual Fear Conditioning

One-Way ANOVA; Factor: Genotype (between-subjects)

|  | % Freezing |
|---|---|

*Table 1 continued on next page*

*Table 1 continued*

| | | |
|---|---|---|
| Genotype | $F_{(3,35)}=5.47$ | $p=0.003$ |
| Morris Water Maze | | |
| Two-Way Factorial ANOVA; Factors: Genotype (between-subjects), Day (within-subjects) | | |
| | Time to platform | |
| Genotype | $F_{(3,80)}=5.17$ | $p=0.01$ |
| Day | $F_{(5,80)}=244.12$ | $p<0.001$ |
| Genotype x Day Interaction | $F_{(5,80)}=1.36$ | $p=0.19$ |

and α2CA1KO mice, but not in α2CA3KO or α2DGKO mice (*Figure 2B*; *Figure 2—figure supplement 1C*; *Table 2*, Sect. 2). The changes (or the lack thereof) in open-arm and lit-compartment activity could not be attributed to nonspecific effects on general locomotor activity, as overall locomotion was not affected by diazepam (*Figure 2C*; *Figure 2—figure supplement 1A*; *Table 2*, Sect. 1, 3). These findings suggest that anxiety-like behaviors are under control of α2GABA$_A$R-mediated inhibition of principal neurons in DG and CA3, whereas a similar level of inhibition of CA1 pyramidal neurons does not affect anxiety-like behavior.

## Behavioral tests of fear

Anxiety and fear are regulated by overlapping but somewhat distinct circuits in the brain (e.g., *Jennings et al., 2013*, *Kheirbek et al., 2013*, *Botta et al., 2015*). Next we tested whether HPC regulation of fear is mediated by an overlapping HPC microcircuit. We used two tests, both of which involve a distinct harmful stimulus (a mild electric shock), to test fear-related behavior.

In the fear-potentiated startle (FPS; *Figure 3A*) test, all experimental groups showed stable baseline startle responses that increased in magnitude with louder white noise bursts during the habituation trials (*Figure 3—figure supplement 1A–D*), showing that a startle magnitude at 85dB (i.e., the magnitude of the testing phase stimulus) is not bound by floor or ceiling effects. On test day, all vehicle-treated mice had increased startle amplitudes when the startle stimulus was preceded by the previously fear-conditioned tone (i.e., FPS; *Figure 3B–E*; *Table 3*, Sect. 1). When GABA$_A$R activity was increased, the magnitude of FPS (%) was reduced significantly in α2F/F control mice. Interestingly, this reduction in% FPS was also observed in α2CA3KO and α2DGKO mice, but not in α2CA1KO mice (*Figure 3F*; *Table 3*, Sect. 1). In a separate test, we evaluated the baseline shock sensitivity of each genotype to the 0.4 mA shock used in FPS, and found no difference between the genotypes (*Figure 3—figure supplement 1E*).

In our FPS test, the fear-reducing effect of diazepam was due not only to a reduction in startle responses in the "tone + startle" condition, but also to an increase in startle magnitude in the "startle stimulus only" condition. While it complicates the interpretation of the data, this effect of diazepam in C57BL/6J mice was documented previously (*Smith et al., 2011*). It should be noted that the startle magnitudes in "tone+startle" condition in diazepam-treated groups was not bound by a ceiling effect, as magnitudes up to 10a.u.'s were observed; thus, the reduction in% FPS in diazepam-treated α2F/F, α2CA3KO and α2DGKO mice is not simply a result of the increased startle in the "startle stimulus only" condition. Still, we conducted a second test of fear, Vogel Conflict Test (VCT), to further clarify these findings.

The genotype-linked contrast between the FPS and the findings from tests of anxiety could be due to different circuitry underlying fear and anxiety, or simply due to different circuitries being involved in unconditioned (EPM and LDB) anxiety versus long-term fear memory. Thus, we chose the VCT, which does not rely on long-term memory (although effects of within-session working memory cannot be eliminated), as our second test of fear. In VCT, water-deprived animals are allowed to drink from a spout which delivers electric shocks to the tongue, creating a conflict between the desire to drink and the desire to avoid a painful stimulus. A reduction in fear of the shock is reflected in increased drinking in the presence of shock. When drinking was unpunished, we found no difference between genotypes (*Figure 3G*; *Table 3*, Sect. 2). In contrast, on test day (punished drinking;

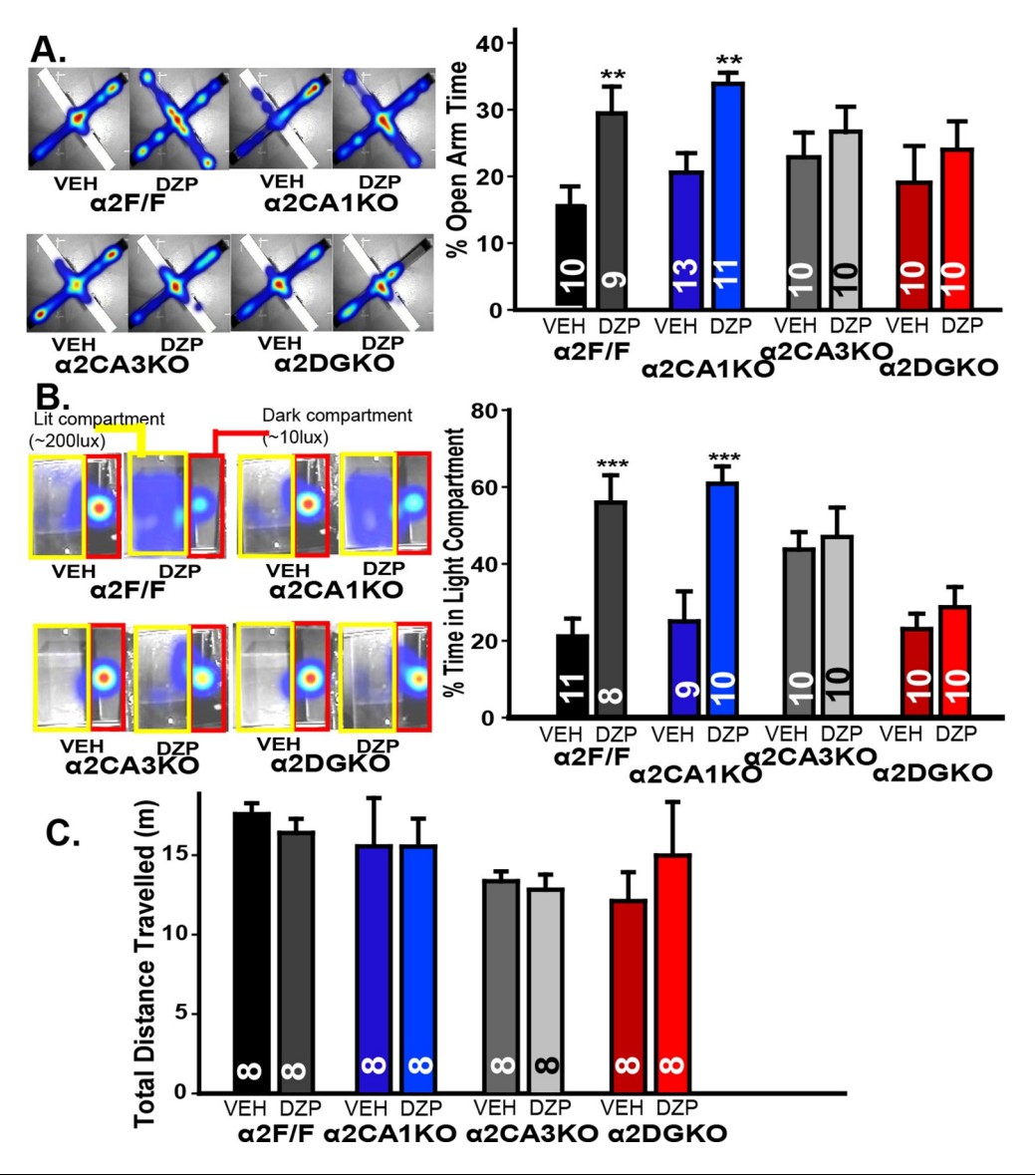

**Figure 2.** Behavioral tests of anxiety and locomotor activity. (**A**) Left: Activity heat maps on the EPM of representative α2F/F, α2CA1KO, α2CA3KO and α2DGKO mice treated with vehicle or diazepam. Right: Percentage (Mean ± S.E.M.) of time spent in the open arms of the EPM. (**B**) Left: Activity heat maps on the LDB of representative α2F/F, α2CA1KO, α2CA3KO and α2DGKO mice treated with vehicle or diazepam. The dark (red) and lit (yellow) compartments of the box are outlined for ease of visualization. Right: Percentage (Mean ± S.E.M.) of time spent in the lit compartment of the LDB. (**C**) Mean ( ± S.E.M.) distance travelled in the open field.

The following source data and figure supplement are available for figure 2:

**Source data 1.** Raw data for elevated plus maze and light/dark box figures.

**Figure supplement 1.** Additional measures in tests of anxiety-like behavior.

*Figure 3H*; *Table 3*, Sect. 2) elevation of GABA$_A$R activity by diazepam significantly increased punished drinking in α2F/F, α2CA3KO and α2DGKO mice. Similar to FPS, the fear-reducing effect of diazepam was abolished in α2CA1KO mice. We also tested the possible effects of diazepam on unpunished drinking and found no nonspecific effects on water consumption (*Figure 3—figure*

**Table 2.** Results of omnibus statistical tests of measured parameters in behavioral tests of anxiety and general locomotion.

### 1. Elevated Plus Maze (C57BL/6J)

Two-Way Factorial ANOVA; Factors: Genotype (between-subjects), Drug (between-subjects)

|  | % Open Arm Time |  | % Open Arm Entries |  | Distance Travelled |  |
|---|---|---|---|---|---|---|
| Genotype | $F_{(3,75)}=1.41$ | p=0.25 | $F_{(3,75)}=0.13$ | p=0.94 | $F_{(3,75)}=0.85$ $F_{(3, 69)} = 1.11$ | p=0.47 |
| Drug | $F_{(1,75)}=16.48$ | p<0.001 | $F_{(3,75)}=2.33$ | p=0.13 | $F_{(3,75)}=0.61$ | p=0.44 |
| Genotype x Drug Interaction | $F_{(3,63)}=1.54$ | p=0.21 | $F_{(3,75)}=1.47$ | p=0.23 | $F_{(3,75)}=1.13$ | p=0.34 |

### 1′. Elevated Plus Maze (129X1/SvJ)

Two-Way Factorial ANOVA; Factors: Genotype (between-subjects), Drug (between-subjects)

|  | % Open Arm Time |  | % Open Arm Entries |  | Distance Travelled |  |
|---|---|---|---|---|---|---|
| Genotype | $F_{(3,55)}=0.17$ | p=0.92 | $F_{(3,55)}=1.59$ | p=0.20 | $F_{(3,55)}=1.17$ | p=0.33 |
| Drug | $F_{(1,55)}=11.49$ | P=0.001 | $F_{(1,55)}=3.4649$ | p=0.07 | $F_{(1,55)}=1.28$ | p=0.29 |
| Genotype x Drug Interaction | $F_{(3,55)}=2.28$ | p=0.09 | $F_{(3,55)}=0.50$ | p=0.69 | $F_{(3,55)}=0.82$ | p=0.49 |

### 2. Light / Dark Box

Two-Way Factorial ANOVA; Factors: Genotype (between-subjects), Drug (between-subjects)

|  | % Time in Light |  | Entries to Light |  |
|---|---|---|---|---|
| Genotype | $F_{(3,73)}=5.03$ | p=0.003 | $F_{(3,73)}=0.84$ | p=0.48 |
| Drug | $(F_{(1,73)}=26.00$ | p<0.001 | $F_{(1,73)}=1.45$ | p=0.23 |
| Genotype x Drug Interaction | $F_{(3,73)}=5.53$ | p=0.002 | $F_{(3,73)}=2.17$ | p=0.09 |

### 3. Open Field

Two-Way Factorial ANOVA; Factors: Genotype (between-subjects), Drug (between-subjects)

|  | Distance Travelled |  |
|---|---|---|
| Genotype | $F_{(3,56)}=1.78$ | p=0.16 |
| Drug | $F_{(1,56)}=0.04$ | p=0.84 |
| Genotype x Drug Interaction | $F_{(3,56)}=0.43$ | p=0.73 |

supplement 1F). Our findings suggest that the double dissociation observed between FPS/VCT and EPM/LDB is likely due to a divergence in the HPC circuitry mediating fear versus anxiety rather than being a consequence of the involvement of memory processes.

## Changes in HPC theta oscillations induced by enhanced GABA$_A$R activity

Previous studies linked theta range oscillations in the HPC to behavioral manifestations of anxiety. For instance, theta activity is enhanced in the ventral HPC (vHPC) with anxiety (*Adhikari et al., 2010*) and pharmacological manipulations that reduce anxiety also reduce the frequency of HPC theta elicited by brain stem stimulation (*McNaughton et al., 1986*, *McNaughton and Coop, 1991*, *Engin et al., 2009*, *Yeung et al., 2012* see also *Wells et al., 2013*). GABAergic anxiolytic drugs additionally reduce the slope of the function that relates brain stem stimulation current to theta frequency (*McNaughton et al., 2007*; *Engin et al., 2008*). As our tests of anxiety indicated that increasing GABA$_A$R function can reduce anxiety in control and α2CA1KO, but not in α2CA3KO and α2DGKO mice, we next tested whether the effect of these manipulations on HPC theta range activity is consistent with the observed behavioral effects. Whereas our genetic manipulations span the whole HPC, previous studies implicated specifically the vHPC in anxiety-related behaviors (*Bannerman et al., 2002*; *Bannerman et al., 2004*; *Fanselow and Dong, 2010*). Thus, we recorded

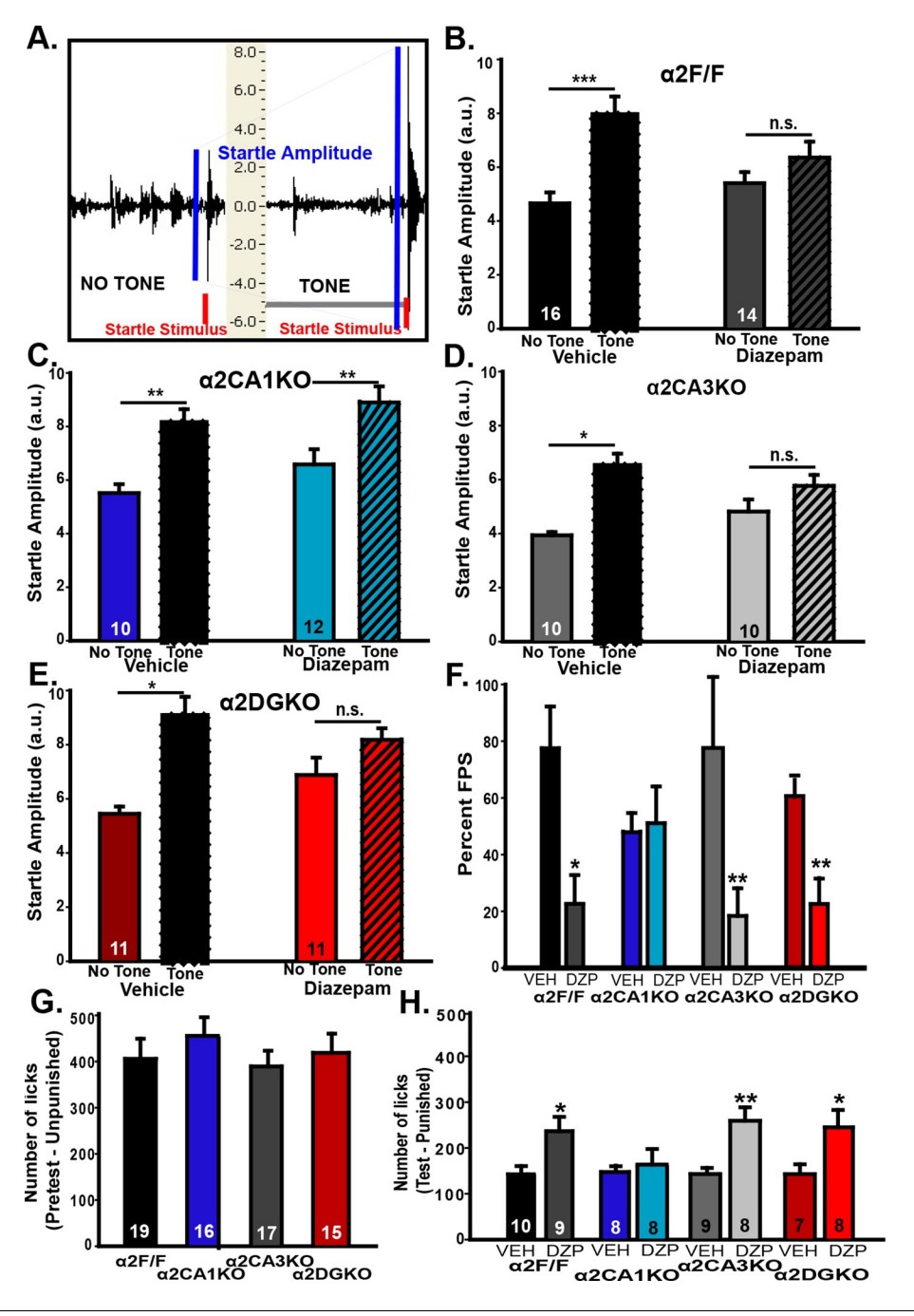

**Figure 3.** Behavioral tests of fear. (**A**) Representative recordings of "Startle Stimulus Only" and "Tone + Startle Stimulus" trials in a α2F/F control mouse treated with vehicle. The increased startle amplitude in "Tone + Startle" trials represents fear-potentiation of the startle response. (**B-E**) Mean ( ± S.E.M.) startle amplitude in "No Tone" and "Tone" startle trials in vehicle and diazepam-treated (**B**) α2F/F, (**C**) αCA1KO, (**D**) α2CA3KO, (**E**) α2DGKO mice. (**F**) Mean ( ± S.E.M.) percent FPS in trials preceded by the tone. Asterisks represent significant difference from the corresponding vehicle group. (**G**) Mean ( ± S.E.M.) number of licks recorded in the pretest session of the VCT where drinking is not punished (This session does not involve drug administration). (**H**) Mean ( ± S.E.M.) number of licks recorded in the test session where every 20th lick is punished in vehicle- or diazepam-treated mice. Asterisks represent significant difference from the corresponding vehicle group. *p<0.05, **p<0.01, ***p<0.001.

*Figure 3 continued on next page*

*Figure 3 continued*

The following source data and figure supplement are available for figure 3:

**Source data 1.** Raw data for fear-potentiated startle and Vogel conflict test figures.
**Figure supplement 1.** Additional measures in tests of fear-related behavior.

separately from the dorsal (dHPC) and vHPC (*Figure 4A*), hypothesizing that the recordings from vHPC might be more closely linked to anxiety.

vHPC recordings indicated that the increase in $GABA_AR$ activity 30 min and 60 min following injection of diazepam reduced the frequency of theta oscillations in the 0.04–0.10 mA stimulation range in both α2F/F and α2CA1KO mice (*Figure 4B,C*; *Table 4*, Sect. 1), in line with the anxiolytic-like behavioral effects of the same drug manipulation in these groups. In α2CA3KO and α2DGKO mice, the peak frequency in theta band was reduced following diazepam, but the reductions were smaller in size and were limited to higher stimulation intensities (*Figure 4D,E*; *Table 4*, Sect. 1). The

**Table 3.** Results of omnibus statistical tests of measured parameters in behavioral tests of fear.

**1. Fear-Potentiated Startle**

Within-Genotype Comparisons

Two-Way Factorial ANOVA; Factors: Tone/No Tone (within-subjects), Drug (between-subjects)

|  | α2F/F |  | α2CA1KO |  |
|---|---|---|---|---|
| Tone | F(1,28)=33.75 | p<0.001 | F(1,20)=49.17 | p<0.001 |
| Drug | F(1,28)=0.30 | p=0.59 | F(1,20)=0.51 | p=0.48 |
| Tone x Drug Interaction | F(1,28)=9.95 | p=0.004 | F(1,20)=0.02 | P=0.89 |
|  | α2CA3KO |  | α2DGKO |  |
| Tone | F(1,18)=16.60 | p<0.001 | F(1,20)=54.46 | p<0.001 |
| Drug | F(1,18)=0.09 | p=0.77 | F(1,20)=0.73 | p=0.40 |
| Tone x Drug Interaction | F(1,18)=5.16 | p=0.04 | F(1,20)=10.24 | p=0.01 |

Between-Genotype Comparisons

Two-Way Factorial ANOVA; Factors: Genotype (between-subjects), Drug (between-subjects)

|  | % FPS |  |
|---|---|---|
| Genotype | F(3,86)=1.17 | p=0.91 |
| Drug | F(1,86)=16.69 | p<0.001 |
| Genotype x Drug Interaction | F(3,86)=2.44 | p=0.07 |

**2. Vogel Conflict Test**

Pretest (Unpunished) Drinking

One-Way ANOVA; Factor: Genotype (between-subjects)

|  | Number of licks |  |
|---|---|---|
| Genotype | F(3,63)=0.63 | p=0.60 |

Test (Punished) Drinking

Two-Way Factorial ANOVA; Factors: Genotype (between-subjects), Drug (between-subjects)

|  | Number of licks |  |
|---|---|---|
| Genotype | F(3,59)=1.00 | p=0.40 |
| Drug | F(1,59)=14.57 | p<0.001 |
| Genotype x Drug Interaction | F(3,59)=1.21 | p=0.31 |

slope of the linear function that relates stimulation intensity to theta frequency was also reduced in all genotypes following diazepam injection (*Figure 4F*), but 30min following injection (the time point at which we conducted our behavioral tests), the magnitude of the reduction in slope was significantly smaller in α2CA3KO and α2DGKO mice compared to controls and α2CA1KOs (*Figure 4F*; *Table 4*, Sect. 2). Thus, the well-validated effects of diazepam on elicited theta activity, which show parallels with anxiolytic behavioral effects, were dampened in α2CA3KO and α2DGKO mice.

Interestingly, simultaneous recordings in the dHPC showed the opposite pattern. Diazepam reduced the frequency of theta in control, α2CA3KO and α2DGKO mice, but there was no main effect of diazepam injection in α2CA1KO mice (*Figure 4—figure supplement 1A–D*; *Table 4*, Sect. 4). The reduction in slope following diazepam injection was also significantly smaller in α2CA1KO mice compared to controls (*Figure 4—figure supplement 1E*).

We found no parallels between behavioral tests of anxiety and fear, and power of evoked theta in dHPC or vHPC (*Figure 4—figure supplement 1F–G*; *Table 4*, Sect. 3, 6), consistent with previous reports (*McNaughton et al., 2007*).

## Discussion

Our findings indicate a double dissociation in the control of anxiety versus fear by HPC microcircuits, with increased α2GABA$_A$R-mediated inhibition of principal neurons in the DG and CA3 nodes of the trisynaptic pathway required for suppression of anxiety, and increased inhibition of CA1 pyramidal neurons required for suppression of fear. Strikingly, as increased GABA$_A$R activity was achieved through a systemic pharmacological manipulation, our findings suggest that even if GABA$_A$R activity is simultaneously increased in all other parts of the circuitry underlying anxiety and/or fear responses, such as the amygdala, bed nucleus of the stria terminalis or the medial prefrontal cortex (mPFC) (*Tovote et al., 2015*) (and in the hippocampus via other GABA$_A$ receptor subtypes), it is not sufficient to suppress anxiety and/or fear responses unless the corresponding HPC subregions are simultaneously inhibited via α2GABA$_A$ receptors.

The findings cannot be explained by changes in expression of the α2GABA$_A$Rs outside of the hippocampus. As seen in *Figure 1* (and *Figure 1—figure supplement 1*), the conditional knockouts were highly specific to the target regions in α2CA3KO and α2DGKO mice. In α2CA1KO mice, there are no significant changes in the expression of the GABA$_A$R α2 subunit protein in the cortex or the amygdala, although there is an apparent change in expression in especially the medial regions of the basolateral amygdala (BLA) in some sections (see *Figure 1*). While this raises the possibility that the effects (or the lack thereof) observed in α2CA1KO mice may be partially due to changes in α2 protein expression in the BLA, our findings from ongoing studies indicate that knocking down α2GABA$_A$Rs in the BLA leads to a completely different pattern of results than that observed in α2CA1KO mice, making this possibility highly unlikely. Our findings also cannot be explained by confounding factors such as gross HPC dysfunction or nonspecific behavioral changes in the gene-targeted mice, as all genotypes showed normal baseline HPC function and response to diazepam, as measured by ex vivo electrophysiology (*Figure 1—figure supplement 3*) and HPC-dependent behavioral tasks (trace and context fear conditioning, Morris water maze; Figure 4—figure supplement 1).

We further report that the underlying mechanism for the suppression of anxiety and fear following the systemic elevation of GABA$_A$R activity may be the effect of this manipulation on vHPC and dHPC theta range activity, respectively. All classes of clinically effective anxiolytic compounds reduce the frequency of evoked HPC theta activity, with no known false positives or false negatives, making this a strong physiological marker of anxiolytic action (*McNaughton et al., 2007*). While most systemic pharmacological manipulations that reduce anxiety also reduce fear, previous studies did not investigate whether the effects of anxiolytic and fear-reducing manipulations on theta range activity may be distinct in this specific model. There is some evidence, however, suggesting that theta range activity recorded from dHPC may be more relevant for fear, while vHPC theta may be more relevant for anxiety. For instance, theta range activity recorded from dHPC in CA1 and the amygdala is synchronized during expression of conditioned fear responses, but not during anxiety-related responses (*Seidenbecher et al., 2003*). Conversely, in recordings from vHPC, increases in theta power and synchrony with mPFC in the theta band were observed during the exploration of anxiogenic arenas, while this was not the case for recordings from the dHPC (*Adhikari et al., 2010*). In a similar vein,

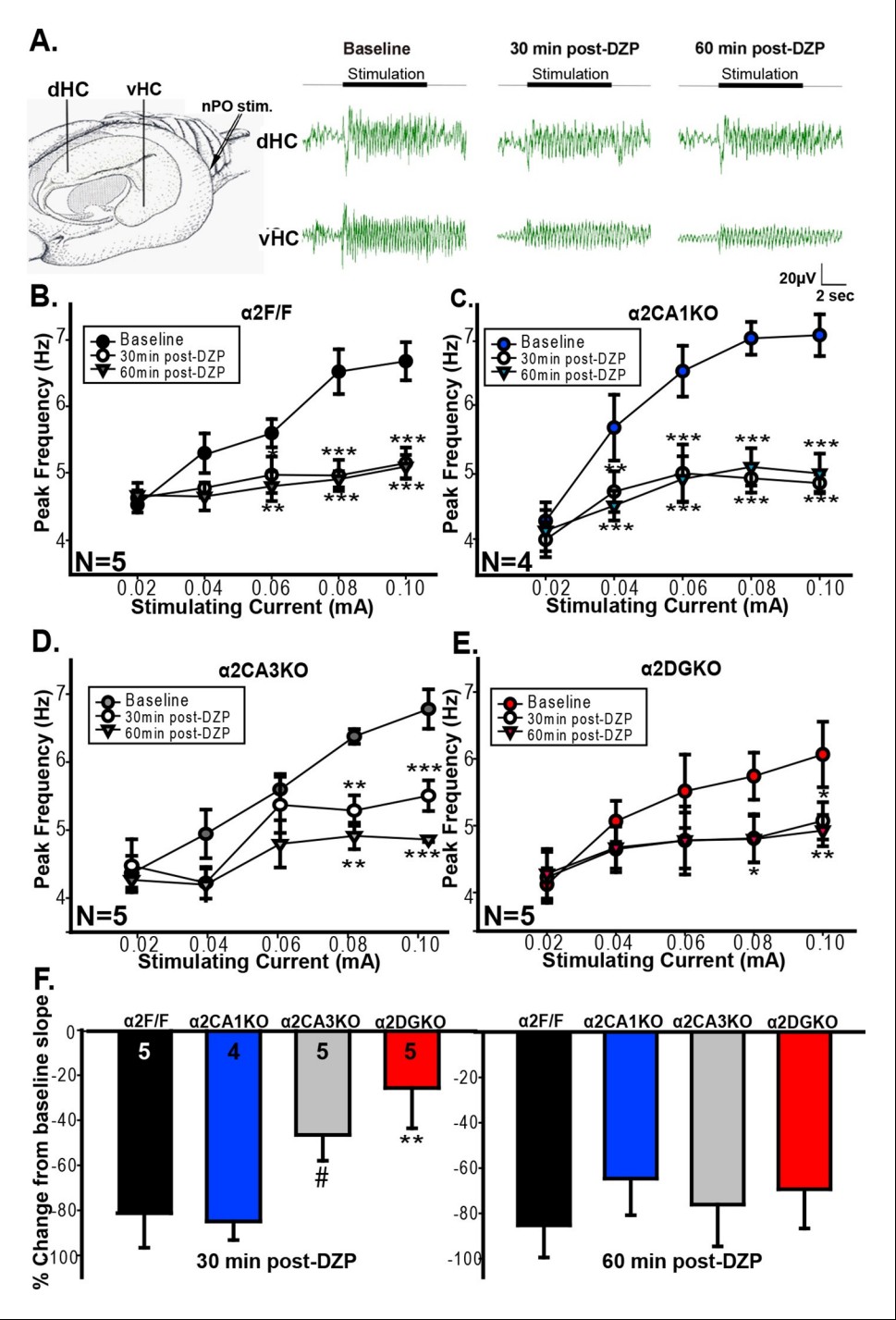

**Figure 4.** Evoked theta oscillations in the vHPC. (**A**) Stimulation and recording sites and representative traces showing vHPC theta activity before and after diazepam injection in a α2F/F mouse. (**B-E**) Mean ( ± S.E.M.) peak frequency in the theta range at different stimulation intensities before (Baseline), and 30 min (30min post-DZP) and 60 min (60min post-DZP) following diazepam injection in (**B**) α2F/F, (**C**) α2CA1KO, (**D**) α2CA3KO, (**E**) α2DGKO mice. Asterisks represent significant difference from the baseline at the given stimulating current, with top ones for 30 min and the lower ones for 60 min post-injection. (**F**) Change from baseline slope in the stimulation intensity – peak frequency function 30 min (left) or 60 min following diazepam injection.#p<0.09, *p<0.05, **p<0.01, ***p<0.001 compared to corresponding α2F/F group.

The following source data and figure supplement are available for figure 4:

*Figure 4 continued on next page*

*Figure 4 continued*

**Source data 1.** Raw data for peak frequency and stimulation intensity – peak frequency slope figures.
**Figure supplement 1.** Frequency and power of theta range oscillations.

we report that diazepam effects on vHPC theta were reduced in α2CA3KO and α2DGKO mice, while effects on dHPC theta were reduced in α2CA1KO mice, in parallel with the effects observed in behavioral tests of anxiety and fear, respectively. While our knockouts mostly span the septo-temporal axis of the hippocampus, these findings also point to the possibility that CA1 in the dHPC may be important for fear, while CA3 and DG in the vHPC may be important for anxiety, integrating our findings with previous reports (*Bannerman et al., 2004*; *Fanselow and Dong, 2010*).

The role of HPC in anxiety and fear was recognized early on (e.g., *Gray, 1982*) and has been demonstrated using different approaches (*Engin and Treit, 2007*; *Bannerman et al., 2004*; *Maren, 2001*). However, despite concentrated efforts in the last 15 years to understand the roles of different HPC subregions in cognitive processes (e.g., (*Nakazawa et al., 2002*; *McHugh et al., 2007*; *Tsien et al., 1996*; *Kesner, 2013a*; *Kesner, 2013b*), and a shift in the focus of anxiety and fear research towards within-structure microcircuits and specific neuronal populations (e.g., *Kim et al., 2013*, *Botta et al., 2015*, *Tye et al., 2011*), a systematic analysis of the regulation of fear and anxiety by HPC microcircuitry had so far not been conducted. Our experiments extend previous research, which to a large extent treated the HPC as a unitary structure except anatomical distinctions along its septo-temporal axis (*Fanselow and Dong, 2010*), and ignored the possible specialization within the microcircuitry of the HPC. Furthermore, the distinction between anxiety and fear, and whether they are processes mediated by distinct neurocircuitry, has been a controversial question (*Perusini and Fanselow, 2015*). Here we show that anxiety and fear, distinguished via experimental conditions that cause each state (we operationally define "tests of fear" as those involving a distinct harmful stimulus; e.g., an electric shock), are mediated by distinct subregions within the HPC, lending further credence to the distinction between the two states and the overlapping but distinct neurobiological underpinnings of fear and anxiety. In conclusion, in addition to demonstrating a surprisingly essential role of the hippocampus in the pharmacological modulation of anxiety and fear, our study indicates distinct molecular mechanisms underlying regulation of two distinct negative valence systems (i.e., anxiety and fear), and provides defined cellular entry points into neuronal circuitry underlying complex behavioral states.

## Materials and methods

All procedures were approved by the McLean Hospital and Yale University Medical School Institutional Animal Care and Use Committees, and were in compliance with the National Research Council Guide for Care and Use of Laboratory Animals (8th Edition, The National Academies Press, Washington, D.C.).

### Animals

For the generation of the floxed *Gabra2* allele, see (*Witschi et al., 2011*) for details. Briefly, a 6.3 kb genomic fragment (PstI-NcoI) containing exons 5 (221 bp) and 6 (83 bp) of the *Gabra2* gene was isolated. A 1 kb SphI-SphI fragment containing Exon 5 was then removed from the 6.3 kb fragment and was replaced by an oligo hybrid containing a *lox*P site in addition to the 1 kb SphI-SphI fragment with exon 5. A neomycine resistance cassette (NEO; FRT-Pol2-neo-bpA-FRT-*lox*P) was subcloned into the SalI site. The vector was electroporated into embryonic stem cells (C57BL6/N, Eurogentec), clones with correctly targeted alleles were injected into blastocysts (Polygene, Rumlang, Switzerland), and the NEO was bred out (Gabra2^tm2.1Uru). *Figure 1A* shows the 6.3 kb PstI-NcoI fragment containing exon 5 (221 bp), flanked by two *lox*P sites, the single FRT site remaining following the excision of the NEO, and exon 6 (83 bp). The α2 conditional knockout mice were generated by crossing mice homozygous for the floxed *Gabra2* allele (*Figure 1A* top; α2F/F mice) with mice that are homozygous for the floxed *Gabra2* allele and carry one of the following three cre recombinase transgenes: CamKIIα cre (T29-1 mice; *Tsien et al., 1996*) to generate a CA1-selective knockout

**Table 4.** Omnibus statistical tests of measured parameters in in vivo LFP recordings collected from ventral and dorsal hippocampus.

**VENTRAL HIPPOCAMPUS**

#### 1. Peak Theta Frequency

Two-Way ANOVA; Factors: Stimulation intensity (within-subjects), Time before/after drug (within-subjects)

|  | α2F/F |  | α2CA1KO |  |
|---|---|---|---|---|
| Stimulation intensity | $F_{(4,32)}=9.13$ | p<0.001 | $F_{(4,24)}=18.12$ | p<0.001 |
| Time after drug | $F_{(2,32)}=22.98$ | p<0.001 | $F_{(2,24)}=37.01$ | p<0.001 |
| Stimulation x Time | $F_{(8, 32)}=4.27$ | p<0.001 | $F_{(8,24)}=9.07$ | p<0.001 |
|  | α2CA3KO |  | α2DGKO |  |
| Stimulation intensity | $F_{(4,32)}=8.14$ | p<0.001 | $F_{(4,32)}=3.84$ | P=0.02 |
| Time after drug | $F_{(2,32)}=6.14$ | P=0.02 | $F_{(2,32)}=2.47$ | P=0.15 |
| Stimulation x Time | $F_{(8, 32)}=3.14$ | P=0.01 | $F_{(8, 32)}=6.63$ | p<0.001 |

#### 2. Stimulation Intensity – Theta Frequency Slope

One-Way ANOVA; Factor: Genotype (between-subjects)

|  | 30min post-diazepam |  | 60min post-diazepam |  |
|---|---|---|---|---|
| Genotype | $F_{(3,15)}=3.94$ | p=0.03 | $F_{(3,73)}=0.84$ | p=0.48 |

#### 3. Normalized Power

One-Way ANOVA; Factor: Genotype (between-subjects)

|  | 60min post-diazepam |  |
|---|---|---|
| Genotype | $F_{(3,15)}=4.65$ | p=0.02 |

**DORSAL HIPPOCAMPUS**

#### 4. Peak Theta Frequency

Two-Way ANOVA; Factors: Stimulation intensity (within-subjects), Time before/after drug (within-subjects)

|  | α2F/F |  | α2CA1KO |  |
|---|---|---|---|---|
| Stimulation intensity | $F_{(4,32)}=8.19$ | p<0.001 | $F_{(4,32)}=5.29$ | p=0.01 |
| Time after drug | $F_{(2,32)}=31.65$ | p<0.001 | $F_{(2,32)}=3.03$ | p<0.11 |
| Stimulation x Time | $F_{(8,32)}=8.87$ | p=0.003 | $F_{(8,32)}=1.89$ | p=0.10 |
|  | α2CA3KO |  | α2DGKO |  |
| Stimulation intensity | $F_{(4,40)}=19.19$ | p<0.001 | $F_{(4,32)}=8.86$ | p<0.001 |
| Time after drug | $F_{(2,40)}=27.08$ | p<0.001 | $F_{(2,32)}=9.25$ | p=0.01 |
| Stimulation x Time | $F_{(8,40)}=4.30$ | p<0.001 | $F_{(8,32)}=2.39$ | p=0.04 |

#### 5. Stimulation Intensity – Theta Frequency Slope

One-Way ANOVA; Factor: Genotype (between-subjects)

|  | 30min post-diazepam |  | 60min post-diazepam |  |
|---|---|---|---|---|
| Genotype | $F_{(3,17)}=2.97$ | p=0.14 | $F_{(3,17)}=4.32$ | p=0.02 |

#### 6. Normalized Power

One-Way ANOVA; Factor: Genotype (between-subjects)

|  | 60min post-diazepam |  |
|---|---|---|
| Genotype | $F_{(3,15)}=1.02$ | p=0.41 |

(α2CA1KO), Grik4 cre (*Nakazawa et al., 2002*) to generate a CA3-selective knockout (α2CA3KO) and POMC cre (*McHugh et al., 2007*) to generate a DG-selective knockout (α2DGKO).

Mice were bred at McLean Hospital animal facility. For immunohistochemistry, ex vivo recordings and FPS experiments, the mice were kept on a 12-hr light/dark cycle with lights on at 07:00 am. For EPM, LDB, open field and VCT, the mice were maintained on a reverse 12-hr light/dark cycle with lights on at 07:00pm, and the tests were conducted during the dark phase. For in vivo electrophysiology experiments, a mix of male and female experimental mice were shipped from McLean Hospital to Yale University at 4 weeks of age, and were allowed to acclimatize to the new environment for at least 6 weeks before the experiments. Only male mice were used for all other experiments. α2F/F mice carrying the corresponding cre transgene were used as experimental animals, while the α2F/F, Cre- littermates were combined into a single α2F/F control group with approximately equal numbers from each breeding. All mice were maintained on either a C57BL/6J background or a 129X1/SvJ (only for the mice used in light/dark box, the secondary elevated plus maze experiments and cognitive measures) background.

## Immunohistochemistry

Mice were deeply anesthetized with sodium pentobarbital (200 mg/kg) and were perfused transcardially with ice-cold phosphate-buffered saline (PBS), followed by 150mM sodium phosphate buffer containing 4% paraformaldehyde and 15% picric acid. The brains were post-fixed in the same fixation solution for 4 hr, and were then processed for antigen retrieval. Briefly, whole brains were incubated in sodium citrate buffer (pH 4.5) overnight at room temperature. Next day, blocks of tissue including the regions of interest were cut, placed in fresh sodium citrate buffer and were irradiated in a microwave for 90 sec. The brains were washed in PBS following microwave irradiation and were placed in 30% sucrose for cryoprotection. The brains were sectioned coronally into 40 μm-thick sections using a sliding microtome, and the sections were stored at -20C in an antifreeze solution until use.

Immunoperoxidase staining was performed using diaminobenzidine as a chromophore. The sections were incubated in a 0.3% $H_2O_2$ solution for 30 min, followed by 2 hr blocking in 3% normal goat serum (NGS), 0.25% Triton X-100 solution for 2 hr, and were incubated overnight at 4C in primary antibodies (Guinea pig anti-α1 (1:20,000), guinea pig anti-α2 (1:1,000) and guinea pig anti-α5 (1:1,000; *Fritschy and Mohler, 1995*). diluted in Tris buffer containing 2% NGS and 0.2% Triton X-100. The next day, the sections were washed and incubated in a biotinylated secondary antibody (goat anti-guinea pig (1:300), Jackson ImmunoResearch), and then in ABC complex solution (Vectastain Elite kit; Vector Laboratories, Burlingame, CA) at room temperature. The sections were incubated in 0.05% diaminobenzidine tetrahydrochloride (Sigma-Aldrich, St. Lois, MO) dissolved in Tris-Triton (pH 7.7) containing 0.02% $H_2O_2$ for approximately 10 min at room temperature, washed in ice-cold PBS, mounted on gelatinized slides. The slides were air-dried, dehydrated and coverslipped with Eukitt (Fluka, Sigma-Aldrich, St. Lois, MO).

For semi-quantitative analysis of the DAB-stained tissue, sections spanning the length of the HPC were photographed at 4x and 10x magnification, and optical density was calculated in regions of interest using the Image J software. For each area, optical density measured on α2F/F control sections was set to 1 and all other measurements were expressed as a proportion of this.

## ROI dissection and RNA preparation

All procedures were carried out in an RNase free environment with surgical tools and bench space decontaminated with RNase AWAY (Molecular Bioproducts, Carlsbad, CA). Hippocampal regions were dissected by laser capture microdissection (LCM). Procedures for LCM were previously described by (*Chen et al., 2014*). Briefly, fresh frozen brains were sectioned at 10 μm and mounted on uncoated glass slides. The slides were then treated with a series of dehydration steps (acetone, ethanol, and xylene) and then air-dried. Hippocampal regions of interest (ROIs), as illustrated in *Figure 1C*, were captured with Arcturus XT (Applied Biosystems, Carlsbad, CA) onto CapSure LCM caps (Applied Biosystems, Carlsbad, CA). For each brain, hippocampal ROIs were collected from 4–5 coronal sections, bilaterally. Total RNA was then purified with PicoPure RNA isolation kit (Applied Biosystems, Carlsbad, CA) from the caps, inspected by Synergy HT (BioTek, Winooski, VT) and normalized to 2 ng/μL concentration. Amygdala and cortical specimens were obtained by manual

dissection. Briefly, after collecting brain sections for LCM, a 300 µm section was cut from the brain and submerged in ice cold PBS. The ROIs, as illustrated in *Figure 1C*, were dissected manually under a magnifying scope. Total RNA was then purified with RNeasy mini kit (Qiagen, Valencia, CA), inspected by Synergy HT (BioTek, Winooski, VT) and normalized to 20ng/µL concentration.

## Reverse transcription and quantitative-PCR

The following primer-sets were used for q-PCR: Gabra2: forward 5′-GCTGCTTCGAATCCAGGA TGA-3′, reverse 5′-AAATCCTCCAGGTGCATGGG-3′; Gabra3: forward 5′-CTTGGGAAGGCAA-GAAGGTA-3′, reverse GGAGCTGCTGGTGTTTTCTT-3′; Gabra4: forward 5′-AAAGCCTCCCCCA-GAAGTT-3′, reverse 5′-CATGTTCAAATTGGCATGTGT-3′. The selectivity of the assays was tested by end-point gel electrophoresis and the primer efficiencies were between 85% and 105%. Reference gene (β-tubulin, forward 5′-GCGCATCAGCGTATACTACAA-3′, reverse 5′-TTCCAAGTCCAC-CAGAATGG-3′) was selected from a pool of 5 candidate reference genes (β-Actin, Cyclin D, HGPRT, S19, and β-tubulin) based on an initial assessment experiment showing its stable expression across strains as well as good selectivity and efficiency. For each RNA specimen, first-strand cDNA was made from 20 ng total RNA by Transcriptor reverse transcriptase kit (Roche Applied Science, Indianapolis, IN). For Gabra2 assay, 1ng cDNA was used in one LightCycler 480 SYBR Green I q-PCR reaction on the LightCycler® 480 Real-Time PCR System (Roche Applied Science, Indianapolis, IN). For Gabra3 and Gabra4 assays, cDNA pre-amplification was performed with equal amount of cDNA input using TaqMan PreAmp Kit (Life Technologies, Grand Island, NY) and q-PCR was subsequently carried out using LightCycler 480 Probes Master (Roche Applied Science, Indianapolis, IN). The $C_T$ values were assessed by the LC480 Software SW1.5 (Roche Applied Science, Indianapolis, IN) and relative expression values were calculated by the $\Delta\Delta C_T$-Method. Statistical analysis was performed by One-Way ANOVA with strain as independent variable, followed by Holm-Sidak *t*-test against α2F/F (control) group. One outlier and 3 failed PCR reactions were excluded from data analysis for Gabra3 and Gabra4 mRNA expression.

## Slice electrophysiology

Vibratome slices of the hippocampus (250–300 mm) were prepared from male α2CA1KO, α2CA3KO or α2DGKO mice or corresponding littermate controls for each group (4 – 7 mice per group). Slices were continuously superfused in solution containing (in mM): 119 NaCl, 2.5 KCl, 2.5 CaCl₂, 1.0 MgSO₄, 1.25 NaH2PO₄, 26.0 NaHCO₃, 10 glucose and equilibrated with 95% O₂ and 5% CO₂ (pH 7.3–7.4) at 22° – 23° C. mIPSC were recorded in CA1, CA3 or DG neurons (i.e., the site of the conditional knockout) in the presence of 10 µM NBQX and 1 µM TTX. Diazepam 1–5 µM was added to the bath solution. Whole-cell recordings of mIPSCs were obtained from pyramidal neurons or granule cells under visual guidance (DIC/infrared optics) with an EPC-9 amplifier and Pulse v8.67 software (HEKA Elektronik). Cells were classified as principal neurons based on spike frequency adaptation in response to prolonged depolarizing current injections. The recording patch electrodes (3–5 MW resistance) contained (in mM): 101.5 K-gluconate, 43.5 KCl, 1 MgCl₂, 0.2 EGTA, 10 HEPES, 2 MgATP, and 0.2 NaGTP (adjusted to pH 7.2 with KOH). Currents were filtered at 1 kHz and digitized at 5 kHz. mIPSCs (recorded in the presence of 1 mM TTX) were analyzed with the Mini Analysis Program v6.0.7 (Synaptosoft Inc.).

## Behavioral tests

### Drugs

Diazepam (BIOMOL International, Plymouth Meeting, PA) was dissolved in a 10% (2-Hydroxypropyl)-β-cyclodextrin vehicle solution (Sigma-Aldrich, St. Lois, MO) through sonication. Diazepam (2 mg/kg, i.p.) or vehicle was administered 30min before the start of the behavioral tests. For in vivo electrophysiological recordings, diazepam was administered s.c. at a dose of 1 mg/kg.

### Fear-conditioning

Separate groups of mice were trained in auditory fear conditioning using a delay or a trace protocol. For the delay protocol, on the first day of the experiment, the mice were placed in a conditioning box (Med-Associates, Inc., St. Albans, VT) with grid floors, and were subjected to 5 tone (20 s, 70 dB, 2800 Hz) - shock (2s, 0.5 mA) pairings with 60 s intervals. The trace protocol was identical

with the exception of a 20 s trace period between the tones and the shocks. 24 h later, the mice were placed in a different context, the tone was played for 6 m and freezing behavior was recorded using the Med-Associates, Inc. Video Freeze Software.

Contextual fear-conditioning was conducted on a separate group of mice. On Day1, the mice were placed in a conditioning box and were given 2 shocks (2s, 1.5 mA, 30 s apart). 24 hr later, they were returned to the same context for 180 s and freezing was recorded.

## Morris water maze (MWM)

The MWM test was conducted in a pool (Diameter: 120 cm) filled with water (22-24°C) made opaque with a white nontoxic dye (Premium Grade Tempera, Blick, Galesburg, IL) containing a submerged escape platform (Diameter: 10 cm). Geometric shapes affixed to the walls served as extra-maze cues. Mice were given 4 trials per day, released from a different quadrant each trial, with the platform location constant. A trial ended either 10 s after the mouse reached the platform, or 60 s after the start of the trial, with the experimenter guiding the mouse to the platform. On probe trials, the platform was removed and the mice were left in the pool for 120 s, and average distance to the platform during this time was measured using the tracking feature of EthoVision XT (Noldus Information Technology, Wageningen, Netherlands). Probe trials were followed by 4 training trials. Following the probe test on Day 12, the platform was moved from the original location to the nearest quadrant (i. e., Reversal) and mice were given the training session with this new platform location from this point on. The mice were placed in a cage with shredded paper towels under a heat lamp until they were dry before being returned to their home cage at the end of testing.

## Elevated plus maze (EPM)

The apparatus was a plus-shaped maze elevated 1 m from the floor consisting of two open (35 cm long × 6 cm wide) and two closed (35 cm long × 6 cm wide × 20 cm high) arms. All testing was conducted under dim lighting (30 lux on open arms). Mice were placed in the center area facing one of the open arms and activity was recorded for 5 min and was scored automatically using the EthoVision XT video tracking system (Noldus Information Technology, Wageningen, Netherlands). The maze was cleaned with 70% ethanol after each animal. Percentage of open arm time ([Open arm time/5 min] × 100) and percentage of open arm entries ([Open arm entries/(Open arm entries + Closed arm entries)] × 100) were used as measures of anxiolysis. Total distance traveled on the maze was used as a within-test measure of general locomotor activity.

## Light / dark box (LDB)

The light / dark box apparatus was comprised of one clear, brightly-lit (200 lux) chamber (28 cm x 28 cm x 31 cm) and a smaller black, dark (<10 lux) chamber (14 cm x 14 cm x 31 cm) connected with a small opening (5 cm on each side). Mice were placed into the dark chamber 30 min. following diazepam injections and were allowed to explore the whole area freely for 6 min. The session was recorded and tracked using EthoVision XT, and total time spent in the lit chamber, as well as the number of entries into the lit chamber were measured automatically. LDB was conducted using mice bred on a 129X1/SvJ background, as a reliable diazepam effect was observed with control mice of this background but not with C57BL/6J background.

## Open field test (OF)

While all other experiments were conducted on naïve mice, the OF test was conducted on the same animals (C57BL/6J) that were previously tested in EPM after a one-week hiatus. Subjects were tested in a clear Plexiglas box (42 cm × 42 cm × 31 cm) evenly illuminated at 100 lux. Subjects were placed in one corner of the box and allowed to explore freely for 30 min. Locomotor activity, measured as the total distance traveled (cm), was analyzed using the EthoVision XT system. Center time measures were not included as a reliable effect of diazepam on this measure was not observed in control mice, supposedly as a result of the relatively non-anxiogenic testing environment due to the familiarity of the testing room.

## Fear-potentiated startle (FPS)

FPS test was conducted with the Med-Associates, Inc. (St. Albans, VT, USA) Startle Reflex System and Advanced Startle software, using a six-day protocol previously optimized for C57BL/6J mice in our laboratory (*Smith et al., 2011*).

On days 1–3 (Habituation Days) animals received 50 semi-random presentations of white noise startle stimuli (20 ms; ten each of 70, 80, 85, 90 and 100 dB) with 30-sec inter-trial intervals. On day 4 (Pre-test Day), animals were first presented with 10 Leader white noise stimuli (20 ms, 85 dB) to habituate to a baseline startle level, and then were presented with 20 "Startle Stimulus Only" (20 ms, 85 dB startle) and 20 "Tone + Startle Stimulus" (30 s, 12 kHz, 70 dB tone followed by a 20 ms, 85 dB startle) trials in a semi-random fashion. On day 5 (Conditioning Day), animals were trained to associate the tone (30 s, 12 kHz, 70 dB) with a footshock (0.25 s, 0.4 mA) through 10 Tone - Shock pairings.

On day 6 (Test Day), mice were assigned to a vehicle or drug group in a systematic-random fashion to assure that the treatment groups were comparable in terms of percent change in startle amplitude on "Tone + Startle Stimulus" trials versus "Startle Stimulus Only" trials based on the Pre-Test Day data. Mice were then injected i.p. 30 min before being placed in the testing chamber for a session identical to that presented on the PreTest Day.

Fear-potentiation of the startle reflex is indicated by a larger startle response "Tone + Startle Stimulus" trials compared to "Startle Stimulus Only" trials on Test Day. Percent FPS was calculated as $(((Tone + Startle) - (Startle Only)/(Startle Only))] \times 100$ using mean startle amplitude values (in arbitrary units) from Test Day.

In a separate test, the mice were placed in Med-Associates, Inc. (St. Albans, VT, USA) fear-conditioning chambers and were given five 0.4 mA shocks to measure possible differences in shock sensitivity between genotypes for the level of shock used in the FPS experiments. The average motion value during shocks (averaged across the five shocks) measured using the Med-Associates, Inc. Video Freeze Software was used as a proxy measure for sensitivity.

## Vogel conflict test (VCT)

VCT was conducted using Med-Associates, Inc. (St. Albans, VT, USA) fear-conditioning chambers fitted with a spout for water delivery on one wall. The experimental procedures were modified from 45, 46, 47, and all tests were conducted on C57Bl/6J background.

On Day 1, in the morning, mice were placed in the testing chamber for 30 min and allowed to explore and drink water from the spout freely (Habituation). The habituation session was repeated in the afternoon of Day 1, and one more time 24 hrs after the initial session on Day 2. Water bottles were removed from home cages following this third habituation session to start water deprivation.

On Day 3 (Pre-test), mice were placed back in the testing chambers for another 30 min session, with the session starting only after the first lick. The number of licks in the first 6 min following the first lick was recorded using custom program on Med-PC Software (Med-Associates, Inc., St. Albans, VT, USA) and the mice to be assigned to drug versus control groups were matched based on this number. Mice were placed back in their home cages without access to water until the test session 24 hrs later. The data from this day was also used to compare drinking between genotypes when drinking was not punished.

On Day 4 (Test), mice were placed in the chambers and the session started with the first lick and continued for 6 min. Mice that did not lick the spout within the first 10 mins were excluded from the test. A 0.5 sec, 0.14 mA shock was administered to the tongue following every 20th lick. Total number of licks was recorded.

In a separate experiment, the above protocol was used with the omission of shocks on Test Day to measure the possible effects of diazepam on unpunished drinking in water-deprived control mice.

## In vivo electrophysiology

Mice were anesthetized with urethane (1.5 g/kg, i.p.) and placed in a Kopf stereotaxic frame (Tujunga, CA) on a temperature-regulated heating pad (Physitemp Instruments Inc., Clifton, NJ) set to maintain body temperature at $37^0$C. Local field potentials were recorded using two concentric stainless steel bipolar electrodes (NE-100X, Rhodes Medical Instruments, Woodland Hills, CA) lowered into the dorsal (−1.94 mm AP, 1.5 mm LM, and -1.4 mm DV) and ventral (−3.16 mm AP,

3.0 mm LM, and -4.2 mm DV) portions of the hippocampus (*Paxinos and Franklin, 2001*). Anterior-posterior and medial-lateral coordinates were measured from bregma, while depth was calculated relative to brain surface. Additionally, a bipolar concentric stimulating electrode (NE-100X, Rhodes Medical Instruments, Woodland Hills, CA) was lowered into the nucleus pontis oralis (nPO; -4.0 mm AP, 1.2 mm ML, and -3.3 mm DV). Local field potentials were amplified using an A-M System (Carlsborg, WA) with filters set between 1 and 500 Hz. The signals were digitized at a rate of 1 kHz, and stored for subsequent analysis using Spike2 software package, version 7 (Cambridge Electronic Design, Cambridge, UK). Electrical stimulation of the nPO consisted of a train of 0.3 ms square pulses delivered over 6 s at a rate of 250 Hz, and was provided by an Isoflex stimulus-isolator (A.M. P.I. Instruments, Jerusalem, Israel). The 6-s stimulation periods were repeated with an interval of 100 s; stimulating current began at 0.02 mA, and increased in 0.02 mA increments with successive stimulations, until a maximum of 0.10 mA was reached. In this way, a stimulus–response was obtained over a total period of 10 min. This pattern of stimulation was repeated without interruption for the duration of each experiment, and the first two stimulation trains were averaged to give baseline values. Fast Fourier transform analysis was performed on the last 5 s of the EEG during each 6-s stimulation period. The first second during stimulation was not included to avoid stimulus artifact (*Scott et al., 2012*; *Siok et al., 2009*). During the analysis, least squares linear regression lines were fitted to the baseline stimulation amplitude – theta frequency data of each mouse separately for ventral and dorsal hippocampal recordings. Animals where increasing stimulation failed to cause a linear frequency response ($R^2$ values < 0.50) were excluded from further analysis (For ventral hippocampus: 2 α2F/F, 1 α2CA3KO, 2 α2DGKO mice; for dorsal hippocampus: 3 α2F/F, 1 α2CA1KO, 1 α2CA3O, 2 α2DGKO mice). As noted above, a mix of male and female mice were used in the analyses (Specifically, 2 female α2F/F and 2 female α2CA1KO mice were used in addition to males due to difficulties in breeding the required number of male mice). While behaviorally there are substantial differences between male and female mice, the hippocampal theta recordings from the female mice did not show any differences from the male and none of the mice that were excluded from the analyses due to not meeting the above-noted criteria were females.

Subsequently to baseline recordings, animals were treated with diazepam (1 mg/kg, s.c.) and the following first to third recordings were averaged to create the 30 min post-injection values, while average of the 4th to 6th recordings as the 60 min post-injection values. Peak theta frequency was measured for each animal by determining where the peak power occurred in the 4–8 Hz frequency band of the power spectrum at a frequency resolution of approximately 0.24 Hz. Stimulating current inducing theta oscillation between 6–7 Hz frequencies was selected for power analysis. Absolute theta power was determined by summing the power in the 4 to 8 Hz frequency band, then normalized for each animal to the mean of the baseline responses prior to drug administration. Changes of theta power from baseline in each group were tested using one-way ANOVA followed by post-hoc Bonferroni's test.

## Statistics

The sample sizes for behavioral tests were calculated with power analyses based on previous findings (*Smith et al., 2012*). The EPM test in the Smith et al. (*Smith et al., 2012*) study uses the same background and test parameters as the EPM test on C57Bl/6J mice in the current study. Thus, the mean and standard deviation values in the control group of the Smith et al. (*Smith et al., 2012*) study were used in a power analysis to calculate sample sizes for all tests of anxiety used in the current study (Mean 1: 0.40, Mean 2: 0.80, Standard deviation values between 0.14–0.18, alpha: 0.05, intended power: 0.80, which yielded target sample sizes of 8–13). Similarly, the FPS test values for control mice in the same study were used to calculate sample sizes for all tests of fear in the current study (Mean 1: 0.45, Mean 2: 0.80, Standard deviation values between 0.24–0.31, alpha: 0.05, intended power: 0.80, which yielded target sample sizes of 7–13). A sample size per group of 4–8 mice was chosen as the group size for in vivo electrophysiology experiments based on earlier studies measuring hippocampal theta oscillations in mice using similar parameters, in which these sample sizes yielded a statistical power of 0.80 or above (*Scott et al., 2012*). Following the exclusion of mice based on criteria explained above (see Materials and methods – In vivo electrophysiology), eventual sample sizes of 3–5 mice were used in statistical analyses.

Data were expressed as means and standard errors of the mean (S.E.M.) and analyzed using the SAS statistical software version 9.1 (SAS Institute, Inc., Cary, NC) and SigmaPlot software version 11.0 (Systat Software, Inc., Chicago, IL).

With the exception of in vitro electrophysiology, data were analyzed with Two-Way Analyses of Variance (ANOVAs) using genotype and drug dose as the factors, followed, where the initial test is statistically significant, by post hoc Holm-Sidak tests for multiple comparisons (unless noted otherwise). In vitro electrophysiology data were analyzed separately for each conditional knockout, with F/F Cre- littermates serving as controls. A two-way ANOVA was used for genotype and drug effects. The significance level for all tests was set at $p < 0.05$.

## Acknowledgements

We thank Konstantin Bakhurin, Lauren Reynolds and Nishani Hewage (McLean Hospital) for assistance in genotyping; Dr. Susumu Tonegawa (MIT/HHMI) for providing CamKIIα-Cre (T29-1) and Grik4- Cre mice, and Dr. Bradford Lowell (Beth Israel Deaconess Medical Center, Boston) for providing POMC-Cre mice. The α2 floxed mice were developed by UR and colleagues and were provided to McLean Hospital by the University of Zurich. The research was supported by the NIMH / NIH Grant MH080006 (to UR) and; Eleanor and Miles Shore Harvard Medical School Fellowship (to EE), Andrew Merrill Memorial Research Fellowship (to EE) and NARSAD Young Investigator Award 19266 (to EE). The Leica TCS-SP8 confocal microscope utilized in this work was purchased with funding from NIH SIG grant 1S10OD010737-01A1. The content is solely the responsibility of the authors and does not necessarily represent the official views of the NIMH or the NIH.

## Additional information

### Competing interests

UR: Received compensation for professional services from Concert Pharmaceuticals in the last three years. The other authors declare that no competing interests exist.

### Funding

| Funder | Grant reference number | Author |
| --- | --- | --- |
| National Institute of Mental Health | MH080006 | Uwe Rudolph |
| Brain and Behavior Research Foundation | 19266 | Elif Engin |
| National Institute of Mental Health | MH095905 | Uwe Rudolph |

The funders had no role in study design, data collection and interpretation, or the decision to submit the work for publication.

### Author contributions

EE, Conception and design, Acquisition of data, Analysis and interpretation of data, Drafting or revising the article; KSS, Conception and design, Acquisition of data, Analysis and interpretation of data; YG, RAF, Acquisition of data, Analysis and interpretation of data, Drafting or revising the article; DN, ET, Acquisition of data, Analysis and interpretation of data; RK, FC, Analysis and interpretation of data, Contributed unpublished essential data or reagents; J-MF, Drafting or revising the article, Contributed unpublished essential data or reagents; VYB, MH, SAH, Analysis and interpretation of data, Drafting or revising the article; UR, Conception and design, Drafting or revising the article

### Author ORCIDs

Elif Engin, http://orcid.org/0000-0002-1804-6811

### Ethics

Animal experimentation: This study was performed in strict accordance with the recommendations in the Guide for the Care and Use of Laboratory Animals of the National Institutes of Health. All of the animals were handled according to approved institutional animal care and use committee (IACUC) protocols (2015N000147, 2014N000263) of McLean Hospital. Every effort was made to minimize suffering.

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
