## [Decision Letter]

Thank you for submitting your work entitled "Modulation of anxiety and fear via distinct intrahippocampal circuits" for consideration by *eLife*. Your article has been reviewed by two peer reviewers, Stephen Maren and Michael Fanselow, and the evaluation has been overseen by Howard Eichenbaum as Reviewing Editor and a Senior Editor.

The reviewers have discussed the reviews with one another and the Reviewing Editor has drafted this decision to help you prepare a revised submission.

Summary:

The reviewers judged that this is an interesting report examining fear and anxiety in several mouse lines with selective knockouts of the GABA_A_R α2 subunit in specific hippocampal subregions, with results suggesting that hippocampal areas CA1 and CA3 and the dentate gyrus make different contributions to fear and anxiety.

Essential revisions:

1) The most robust loss of diazepam sensitivity on tests of anxiety (e.g., LDB) were performed in mice bred on a 129X1/SvJ background. This line of mice has been reported to show a high-anxiety phenotype (see Camp et al. 2009, Genes Brain Behav), which likely accounts for the lower levels of light-box time in this strain. Indeed, the differences in fear conditioning and extinction between C57 and 129S1 and 129X1 mice is well characterized. That said, the major concern is that all the measures (e.g., basal behavior, immunohistochemistry, RNA, in vivo and in vitro electrophysiology) used to establish the nature and pattern of α2 KO were performed in C57s (as near as I can tell). Of course, all of these parameters might differ markedly in the 129X1 animals (including the specificity for the KO).

2) In Figure 1, immunohistochemical evidence is reported to support the claim that the KOs exhibited selective patters of α2 KO. These data appear to indicate that amygdala protein expression is lower in the CA1 and CA3 KO (quite dramatic in Figure 1, and apparent in the quantification in Figure 1). Although this difference apparently was not statistically significant, it nonetheless appears pronounced and raises questions about how α2 KO in the amygdala either alone or in combination with the hippocampal KOs contributes to the observed effects.

---

## [Author Response]

1) The most robust loss of diazepam sensitivity on tests of anxiety (e.g., LDB) were performed in mice bred on a 129X1/SvJ background. This line of mice has been reported to show a high-anxiety phenotype (see Camp et al. 2009, Genes Brain Behav), which likely accounts for the lower levels of light-box time in this strain. Indeed, the differences in fear conditioning and extinction between C57 and 129S1 and 129X1 mice is well characterized. That said, the major concern is that all the measures (e.g., basal behavior, immunohistochemistry, RNA, in vivo and in vitro electrophysiology) used to establish the nature and pattern of α

*2 KO were performed in C57s (as near as I can tell). Of course, all of these parameters might differ markedly in the 129X1 animals (including the specificity for the KO).*

We agree with the reviewers that certain differences in behavior may be due to strain differences and may indeed explain the fact that we were not able to get a reliable diazepam effect in LDB in C57BL/6J mice, while this was possible in 129X1/SvJ mice. There is also reason to believe that α2 expression levels may be different between the two strains e.g., SD Schlussman et al., 2013 Brain Res 1523: 49-58 shows that a different 129 strain, 129P3/J, has higher α2 mRNA expression compared to C57BL/6J) Other studies (MK Mulligan et al., 2012 PLoS ONE 7: e34586; R.W. Overall et al., 2009 Front Neurosci 3: 55) indicated that the C57BL/6J mice had the lowest level of α2 mRNA expression in the hippocampus out of 99 studied inbred strains. As repeating all experiments in the 129X1/SvJ strain is not practically possible, we have now included immunostaining for α2 subunit with the conditional knockouts on the 129X1/SvJ background to at least demonstrate the pattern of knockouts in these mice (Figure 1—figure supplement 1). Please note that as the immunohistochemical procedures were not carried out in parallel, it is not possible to make direct comparisons of staining intensity between the sections shown in Figure 1 and those in Figure 1—figure supplement 1. Thus, any apparent differences between C57 and 129 sections in terms of overall staining intensity in these figures are arbitrary. However, all sections shown in Figure 1—figure supplement 1 were processed in parallel and can be directly compared to each other. As seen, the pattern of knockdowns is highly similar to those observed in C57BL/6J animals. This is now also noted in the text (Results; Behavioral Tests of Anxiety). The only test that was carried out in the 129X1/SvJ strain was the LDB, while EPM was conducted in both strains in separate experiments, with comparable results.

FPS, VCT and in vivo electrophysiology were all conducted in C57BL/6J mice.

2) In Figure 1, immunohistochemical evidence is reported to support the claim that the KOs exhibited selective patters of α2 KO. These data appear to indicate that amygdala protein expression is lower in the CA1 and CA3 KO (quite dramatic in Figure 1, and apparent in the quantification in Figure 1). Although this difference apparently was not statistically significant, it nonetheless appears pronounced and raises questions about how α2 KO in the amygdala either alone or in combination with the hippocampal KOs contributes to the observed effects.

We now discuss this possibility briefly in paragraph 2 of the Discussion. It should be noted that the quantification of neither the immunohistochemistry nor the RT-PCR data indicates a significant reduction in the amygdala, making it unlikely that the effects observed in α2CA1KO mice are due to expression changes in the amygdala. Furthermore, in an ongoing study, we are investigating the effects of a basolateral amygdala selective knockdown of α2GABA_A_Rs in anxiety and fear-related behaviors. The pattern of effects observed when α2GABA_A_R expression is significantly reduced in the BLA is very distinct from those observed in α2CA1KO mice (e.g., lack of a diazepam effect in EPM; baseline% FPS significantly reduced to almost zero). As these studies are ongoing, we are unfortunately unable to report the results here, but we have mentioned that these studies reveal a distinct pattern of effects in our Discussion per the suggestion of the editors.